# A protocol for using rapid qualitative techniques to incorporate multi-level stakeholder feedback in a pragmatic clinical trial of mindfulness for chronic low back pain

Isabel J. Roth[1]*, Elondra D. Harr[2], Christine R. Lathren[2], Jessica L. Barnhill[2], Ruth D. Rodriguez[3], Jose E. Baez[4], Natalia E. Morone[4]

1 University of North Carolina at Chapel Hill School of Nursing, Chapel Hill, North Carolina, United States of America, 2 Program on Integrative Medicine, Department of Physical Medicine and Rehabilitation, University of North Carolina at Chapel Hill School of Medicine, Chapel Hill, North Carolina, United States of America, 3 Boston Medical Center, Boston, Massachusetts, United States of America, 4 Boston University Chobanian & Avedisian School of Medicine, Boston Medical Center, Boston, Massachusetts, United States of America

* iroth@email.unc.edu

## Abstract

### Introduction

Engaging community members and context experts is increasingly recognized as key to developing research that is responsive to community needs. Here, we describe a protocol for incorporating stakeholder feedback using rapid qualitative techniques into OPTIMUM (Optimizing Pain Treatment In Medical settings Using Mindfulness), a pragmatic clinical trial comparing a telemedicine-delivered mindfulness-based stress reduction intervention to usual care to address chronic low back pain. The aim of this stakeholder feedback supplement to the OPTIMUM parent trial is to consider many viewpoints regarding recruitment, retention, facilitation, delivery, sustainability, and dissemination of this program which are critical to understand before it can be successfully implemented.

### Methods

Our team developed a multi-faceted approach to collecting feedback from representatives of three levels of influence: individuals, communities, and policy. We plan to conduct focus groups with study participants from both the intervention (MBSR) and usual care groups. We plan to conduct one-time semi-structured interviews with a diverse set of people with varied roles and perspectives (e.g., clinic personnel, health care system leadership, mindfulness instructors, patient pain advocacy groups, policy advocates). We will assemble a Community Advisory Board (CAB) to convene regularly throughout the project. Transcripts from interviews, focus groups, and meeting

**Data availability statement:** Data will be de-identified and stored in a data repository to be approved by the National Institutes of Health. Data will be made available when all main study procedures have been completed and the main outcomes paper has been published.

**Funding:** This work was supported within the National Institutes of Health (NIH) Pragmatic Trials Collaboratory through the NIH HEAL Initiative under award number UH3AT010621 administered by the National Center for Complementary and Integrative Health (NCCIH). This work also received logistical and technical support from the PRISM Resource Coordinating Center under award number U24AT010961 from the NIH through the NIH HEAL Initiative. The funders had no role in study design, data collection and analysis, decision to publish, or preparation of the manuscript.

**Competing interests:** The authors have declared that no competing interests exist.

notes will be analyzed using rapid qualitative methods to facilitate timely incorporation of feedback into the trial. In-depth thematic content analysis will be conducted subsequently.

## Discussion

Partnering with communities who are historically underrepresented in clinical research under the guidance of principles such as equity, inclusion, trust, and accountability can improve health outcomes that are most relevant and beneficial to the target community, accelerate uptake, and promote sustainability.

## Introduction

Engaging community members and context experts is increasingly recognized as key to developing research that is responsive to community needs. Working with representatives from the community where the research is taking place, getting feedback from individuals with lived experience with the condition being studied, and collaborating with individuals from groups who will eventually implement interventions under study can provide critical context to research teams. Principles of Community-Based Participatory Research, in which context experts are engaged throughout the research process to promote greater sustainability of interventions as well as community acceptance of those interventions, can guide this process [1]. Of note, while we at times refer to this work collectively as "stakeholder engagement", we recognize that the term "stakeholder" has come under criticism for obscuring the broad range of persons, entities, and interests that this term can encompass [2]. Therefore, when possible, we describe the types of community members and context experts that we will recruit through our stakeholder engagement protocol.

Literature on stakeholder engagement processes and outcomes is emergent and growing. In recent years, efforts to consolidate knowledge into conceptual frameworks and best practices for planning, conducting, and evaluating stakeholder-engaged work have been described [3–5]. Although often challenging [6] best practice guidelines often include the importance of being responsive to stakeholder input [7]. Using rapid qualitative techniques can shorten the typically lengthy time it takes to analyze qualitative data and thus help research teams synthesize stakeholder feedback and use it to refine protocols, optimize recruitment, maximize retention, and prepare for future implementation and dissemination in real-time [8–11]. However, literature on how this is accomplished within the context of an ongoing trial is sparse.

Here, we describe a protocol for incorporating stakeholder feedback using rapid qualitative techniques into OPTIMUM (Optimizing Pain Treatment In Medical settings Using Mindfulness), a pragmatic clinical trial (PCT) comparing a telemedicine-delivered mindfulness-based stress reduction (MBSR) intervention to usual care to address chronic low back pain (cLBP). MBSR is a mind and body therapy that is now included in the evidence-based guidelines of the American College of Physicians for initial treatment of cLBP and it is a Best Practice recommendation of the U.S.

Department of Health and Human Services [3,4]. Yet despite its effectiveness at decreasing pain and improving function, MBSR remains underutilized and has not been incorporated into routine clinical care. Likely barriers to its integration include lack of procedural guidelines to deliver a group program in the clinic, unfamiliarity with MBSR, and uncertain reimbursement. OPTIMUM is being conducted to address these barriers and determine whether MBSR is effective in real-life settings. The OPTIMUM protocol, including details about the MBSR intervention is available for reference [5].

We developed a stakeholder engagement protocol that would be enacted simultaneously with the OPTIMUM trial protocol to address three primary goals: (1) to refine protocols for the recruitment and retention of study participants; (2) to inform the future direction of this research, and (3) to facilitate dissemination of study findings. This protocol is an example of using rapid qualitative analysis to put stakeholder input into practice in real-time in an ongoing clinical trial. This protocol was developed after initiation of the OPTIMUM trial when funding for stakeholder engagement became available through a supplemental award, and is guided by the Social-Ecological Model [6]. The model recognizes that multiple factors influence health behaviors, including factors at the individual, community, and policy levels. In addressing the uptake of MBSR within a primary care context, the Social-Ecological Model helps to illustrate that multi-level stakeholders (i.e.,; representing individuals, communities, and advocates) will impact adoption of MBSR and thus will have perspectives that are critical to consider for successful implementation.

## Materials and methods

**Summary of Parent Study:** OPTIMUM is a randomized pragmatic trial being conducted in three health care systems (Boston Medical Center, MA, UPMC, Pittsburgh, PA, and Piedmont Health Services, in partnership with the University of North Carolina (UNC) Chapel Hill). These three systems represent a range of primary care clinic types and patients served: Boston Medical Center is a safety-net institution serving an urban, low-income, racially and ethnically diverse population. UNC Chapel Hill has two sites: an academic center-based family medicine clinic, and a federally qualified health center serving many rural patients and those underrepresented in research. University of Pittsburgh serves an urban population. Four-hundred-fifty persons (approximately 150 from each site) with cLBP ≥ 18 years will be randomized to 1) OPTIMUM (n = 225) + Primary Care Provider (PCP) Usual Care; or 2) PCP Usual Care (n = 225). Our primary hypothesis is that patients in the MBSR group will have less pain intensity and pain interference at 6-months (primary endpoint) and at 12-months, compared to study participants receiving PCP Usual Care. See Fig 1 for an overview of the study design.

The OPTIMUM study is integrating an evidence-based group MBSR program for chronic low back pain into primary care via telehealth. There are significant barriers to delivering group MBSR, as primary care is traditionally delivered one-on-one (one provider-one patient, in person). For this reason, we are conducting a PCT to determine if MBSR delivered in primary care under the usual clinical workflows will result in a reduction of chronic pain and increase physical and psychological function. Because OPTIMUM will be delivered within the clinical setting, there are many viewpoints to consider regarding recruitment, retention, facilitation, delivery, sustainability, and dissemination of this program which are critical to understand before it can be successfully implemented. Our team developed a multi-faceted approach to collecting feedback from representatives of three levels of influence: individuals, communities, and policy. At the individual level, we plan to conduct focus groups with study participants from both the intervention (MBSR) and usual care groups. Representing individual, community, and policy change advocates s, we plan to conduct one-time semi-structured interviews with a diverse set of people with varied roles and perspectives (e.g., clinic personnel, health care system leadership, mindfulness instructors, patient pain advocacy groups, policy advocates). Finally, also representing a mix of all three levels, we will assemble a Community Advisory Board (CAB) to convene regularly throughout the project. Details on these three modalities are provided below.

**Focus Groups:** The purpose of study participant focus groups is to provide context on factors that influenced their decision to enroll, potential hurdles to enrollment, and issues that would impact or facilitate retention within the trial. Focus groups will provide context and suggest strategies to directly improve recruitment and retention efforts that are not

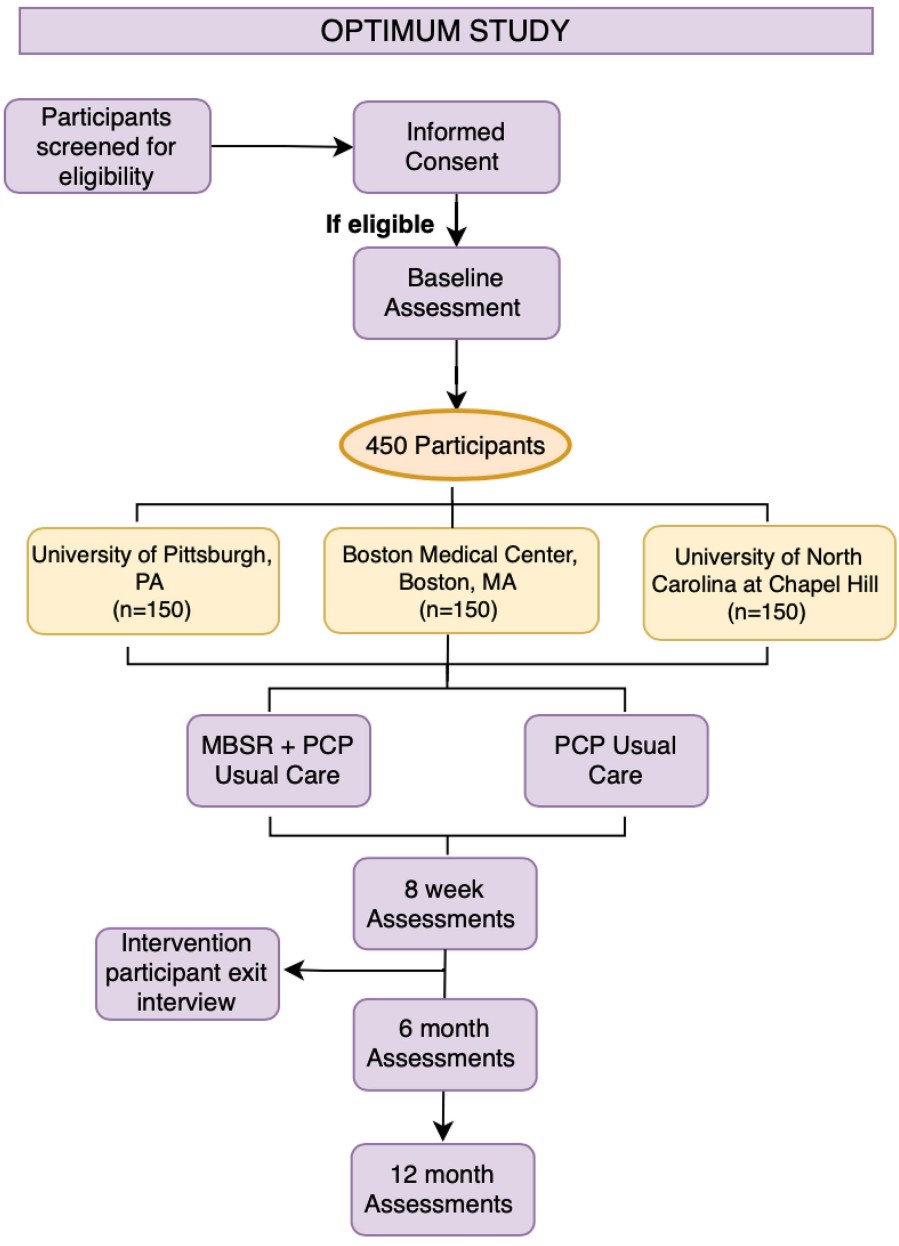

**Fig 1. The OPTIMUM Clinical trial, the parent study to this protocol, involves screening patients for eligibility, if eligible, patient complete informed consent procedures and complete a baseline assessment.** One hundred and fifty patients are recruited from each of the three participating sites, University of Pittsburgh, PA, Boston Medical Center, Boston, MA, and University of Noth Carolina at Chapel Hill, for a total of 450 participants. Participants are then randomized to either receive usual care, or usual care with Mindfulness-Based Stress Reduction (MBSR), which is an 8-week long intervention. Participants complete assessments at 8-weeks, 6 months, and 12 months. Participants in the intervention group are invited to participate in an exit interview after completing the MBSR intervention at 8 weeks.

currently used in the parent study. We will conduct focus groups with the intervention group as well as the control group at each of the three research sites. To accommodate the perspectives of both intervention and usual care groups, the interview guide will contain separate sections of questions that are specific to both groups (see Appendix B in S1 Appendix).

Participants in the OPTIMUM trial who have signed informed consent will be invited to participate in virtual focus groups after they have been in the trial for at least 8-weeks. By this time point we expect that they will be familiar with trial procedures and will be able to provide informed feedback. Focus groups will be conducted at each OPTIMUM clinical site every three months during the first 18 months of recruitment and facilitated by study staff. The focus groups will be supervised by a moderator (IR, NM, CL, JB) who will direct the conversation based on a list of focused questions that aim to elicit several points of interest. Approximately six focus groups will meet for comprehensive 120-minute discussions about their perspectives regarding participation in MBSR for cLBP in primary care via telehealth or being a part of the comparison arm. Approximately eight people will participate in each focus group and participants will be reimbursed for their time. Participants will complete a consent process prior to the focus group. We are conservative in choosing to conduct six focus groups as qualitative methodology literature states thematic saturation can be achieved with 3–5 focus groups [12]. If thematic saturation is not reached within the proposed six focus groups, we will conduct additional focus groups with as needed. The audio for each session will be recorded and the discussions transcribed and then modified to ensure sensitive information regarding identity is removed.

To ensure that a diverse sample of underrepresented patients are represented in the focus groups in the supplement study, we will first evaluate the reported demographic factors of all potential participants. A random number generator will be used to select 10 individuals (assuming that there will be some drop-off to reach our desired group size of 8 participants), followed by an assessment to ensure that the group reflects the diversity of the parent study population. Participants will be assigned a number attached to their participant ID and the generator will randomly select a number. The research team will then contact the participant to invite them to participate and if a selected participant is unavailable, another demographically similar participant was contacted to maintain group diversity.

**Multi-Level Stakeholder Interviews:** The purpose of multi-level stakeholder interviews is to explore the feasibility of delivering MBSR to cLBP patients in primary care in a sustainable fashion. To include a range of perspectives on recruitment, retention, integration, and dissemination, we will interview health system personnel (such as providers, nurses, support staff), pain advocacy groups, mindfulness organizations, healthcare administrators and payors. Stakeholders will be recruited from the existing OPTIMUM clinical sites, as well as upper leadership from these health care systems. Representatives from local insurers (including state Medicaid representatives and private health insurers) will be recruited to participate in interviews. We will use snowball sampling to identify additional stakeholders through these interviews by directly asking about other important people to interview. Multi-level stakeholder interviews will occur throughout the duration of the OPTIMUM study and will follow a menu of questions tailored for the type of interviewee, while also being semi-structured to allow for flexibility in discussion. Interviewees will be invited via e-mailed invitations and interviews will occur virtually to allow audio recording and subsequent transcription. All interviewees will complete a consent process prior to the interview and will be reimbursed for their time. Interviews will continue until thematic saturation is reached.

**Rapid Qualitative Methods to Incorporate Interview/Focus Group Feedback:** The transcripts will be analyzed on an ongoing basis as focus groups and interviews occur by OPTIMUM investigators (IR, NM, CL, JB). We will utilize the Lighting Report method to summarize meetings and share feedback with the research team [7]. With this method, real time synthesis of the CAB's discussion occurs under classifications of "what works" (Plus), "what needs change" (Delta), and "insights, ideas or recommendations" (Insights). Semi-structured interviews will be rapidly analyzed using the summary methodology, highlighting important topics and action items [13]. The summary method is a more flexible format than Lightning Reports, that will allow us to capture the diversity of topics covered in the multi-level stakeholder interviews. The research team will present to the CAB what recommendations can or cannot be implemented and why or why not. The Lightning Reports will be brought to the CAB as well as the OPTIMUM team at all three health care systems at three-month intervals. Rapid qualitative techniques will also allow the analysis team to determine when thematic saturation has been reached in a timely fashion, as analysis will be conducted as data is being collected.

**Transcript Analysis:** Following rapid analysis, transcripts will also be coded in-depth. Codebook development, line-by-line coding, and thematic analysis will be performed on the focus group and interview transcripts by study members using a combination of editing and template organizing styles. This detailed review of the transcripts allows us to add any reflections or recommendations missed in the Lightning Reports and Summaries which are generated in real-time. We will create codes and a codebook and then go through the transcript using these codes to organize and analyze the data (template style). We will also go through the transcript line by line (editing style) and add or change codes from the original codebook. The codebook template is developed through an iterative editing process. Once the codebook is agreed upon, the coders will each code all transcripts using ATLAS.ti qualitative data management software [14], then they will meet to adjudicate all coding differences. Following coding, quotes associated with various themes will be reviewed using the constant comparative method to determine similarities and differences between the way focus group participants approached each topic that was coded [12].

**Community Advisory Board:** The goal of the Community Advisory Board (CAB) is to have a significant and lasting impact on the ability of the OPTIMUM study to recruit and retain a diverse patient population as well as address implementation and dissemination. The CAB will include 10 members and will be comprised of 5 patients who have cLBP, 1–2 mid-level administrators, 1–2 representatives from pain advocacy groups, 1 representative from a mindfulness organization, and 1 behavioral health professional. The CAB will meet monthly over Zoom, at a time most convenient for all participants, and will be facilitated by research team members trained in Community-Based Participatory Research.

The CAB's responsibilities include giving meaningful feedback in three strategic engagement areas: 1) participant recruitment and retention, including recruitment materials and methods, as well as retention efforts, and as the study progresses, dissemination, 2) provide perspectives on barriers to integration of MBSR into primary care and suggest additional strategies for integration, and 3) review summaries of trial progress to facilitate transparency and accountability.

The CAB will undergo in-depth training on how to be a stakeholder in the research process before the monthly meetings occur. We will use the Connecting Community to Research: A Toolkit training for community-engaged research developed under the auspices of the Patient-Centered Outcomes Research Institute [15]. This 3-hour training will set expectations for how the CAB will operate, including building rapport within the group, setting ground-rules, and clarifying roles and responsibilities of board members. An introduction to the OPTIMUM research project, including an introduction to mindfulness, introduction to group medical visits, and an overview of each study site and team will also be reviewed. We will seek feedback from CAB members regarding the training to assess its appropriateness and effectiveness through the development of an online survey that will include open-ended questions such as: 1) What did you like about today's training? 2) What didn't work for you? (i.e., timing, format, content, etc.) 3) How would you improve today's meeting/training? And 4) What would you like us to keep doing in future CAB meetings and/or trainings? Additionally, we will use the Give/Get Model during the training by having each patient and stakeholder describe their interests and expectations (what will they give/what will they get) from their CAB participation [16].

We will offer additional trainings to the CAB to hone their expertise and respond to requests for additional topics (e.g., research ethics). Following this in-depth training, CAB members will meet monthly via videoconference. Board members will be compensated for their participation. Topics to be covered are reviewed in Table 1.

**Rapid Qualitative Methods to Incorporate CAB feedback.** We will also use the Lightning Report method described above during the CAB meetings to increase the accuracy and validity of the stakeholder data and to create reports that can be shared in a timely manner with the CAB and the research team. The synthesis is discussed towards the end of the meeting with the CAB to increase the accuracy and validity of the points discussed. These points will then be brought to the study investigators who will need to respond to the recommendations made. The response will be operationalized through a grid that lists each discussion point and the research team's response. The response will include how the recommendation will be implemented or if not, why not. We expect that the diverse make-up of the CAB will provide

**Table 1. Topics to be covered during monthly CAB meetings include.**

| |
|---|
| Overview of recruitment processes, current barriers and facilitators to recruitment, solicitation of feedback on best practices for recruitment |
| Discussion of best practices for outreach to diverse and under-represented patient populations |
| Regular updates on recruitment, retention, and solicitation of feedback |
| 1. Discussion of barriers and facilitators to implementation of MBSR and GMVs in primary care settings |
| 2. Brainstorm potential strategies to facilitate implementation of OPTIMUM in primary care settings |
| 3. Discussion of trainings needed for facilitators of OPTIMUM (MBSR instructors and providers) |
| 4. Discussion with research team of OPTIMUM study findings, interpretation |
| 5. Policy implications of OPTIMUM findings |
| 6. Dissemination strategies for OPTIMUM study findings |

MBSR = mindfulness-based stress reduction; GMV = group medical visit.

the OPTIMUM study with real-time feedback, recommendations, and insights into all areas of engagement (recruitment, retention, dissemination).

**Data Management Plan:** All data will be transcribed by an encrypted transcription service. Data will be stored using an encrypted cloud server that will be made available to all investigators. Transcripts will be compiled into project files for analysis in ATLAS.ti qualitative data management software and maintained by a research assistant.

**Study Status and Timeline:** The stakeholder engagement supplement to the OPTIMUM trial was funded in September of 2021. Recruitment and data collection activities began on May 14th, 2022, remain ongoing, and are planned to continue through September 1st, 2024, when data collection for the parent trial is expected to be complete. All research participants (focus group and interview participants) provided verbal informed consent. CAB members are not considered research participants.

## Discussion

This protocol paper provides our strategy for engaging multi-level community members and context experts in multiple ways and incorporating their values, opinions and insights in real-time using rapid qualitative techniques. These techniques include Lightning Reports to translate these reports to action items, and procedures to communicate and coordinate with the larger team and stakeholder constituents. We anticipate that being timely, responsive to, and transparent regarding implementation of stakeholder suggestions will foster and reinforce trusting relationships between the research team and our expert advisors and will contribute to the success of this project.

Although community engagement is relevant to most any patient-centered work, meaningful stakeholder engagement activities are particularly important for research that aims to decrease health disparities and improve outcomes because these activities center the voices of those the research strives to benefit. Partnering with communities who are historically underrepresented in clinical research under the guidance of principles such as equity, inclusion, trust, and accountability can improve health outcomes that are most relevant and beneficial to the target community, accelerate uptake, and promote sustainability. However, initiating and maintaining authentic engagement through relationships is challenging and requires significant commitment and resources [16,17]. Specifically, we may not obtain adequate representation of underrepresented groups in our sample from patients and other key stakeholders in this study. To address this potential limitation, we may consider using various strategies to increase the diversity of our samples. For our patient sample, we may utilize sampling methods such as purposive sampling to ensure that we receive a more accurate representation of underrepresented groups in our focus groups. Additionally, because we are unable to contact participants who declined to participate in the parent trial (our IRB requires that we delete their contact information after they have declined participation),

we will miss out on the perspectives of those who may have faced barriers to participation. While snowball sampling may be the most appropriate sampling method for developing the CAB and conducting multi-level stakeholder interviews, we can increase diverse representation by utilizing our existing connections and networks to ask for assistance in increasing diversity and representation in our sample. We may also engage community organizations and advocacy groups to establish connections with underrepresented populations. In doing so, these strategies will help us target specific underrepresented groups in both the parent study and the stakeholder supplement, increasing the likelihood of reaching thematic saturation.

In recent years, there has been an increase in community involvement in research [18], yet there is still a need for more research on how to implement stakeholder engagement results into ongoing pragmatic research studies. We believe that this collaboration between multilevel stakeholders will sow the seeds for future phases of research, where key takeaways from this process can be implemented on a larger scale and integrated into existing resources for pragmatic trial researchers.

Additionally, the primary outcome of interest in this study—improving function and decreasing pain interference for those with chronic low back pain—is one where lived experience provides important context for treatment. Chronic pain is particularly impactful in low income and racially and ethnically diverse patients whose access to treatment is often limited [19]. However, there are significant barriers to accessing treatment among these populations such as limited health insurance coverage, lack of access to integrative health therapies through their providers, and structural barriers such as transportation issues, low socioeconomic status, low health literacy, and racism [19–23] OPTIMUM utilizes a virtual format to circumvent some of the barriers to accessing integrative care. Previous findings suggest that group medical visits have led to an increase in health-related quality of life, patient satisfaction and patient trust in their physician as well as reduced patient care costs [24–26]. By providing a virtual group mindfulness pain program in primary care, we hope to improve access to the program, and thus, to improve health. An innovative aspect of the OPTIMUM study is that we will be able to address questions related to offering mindfulness interventions to low income and racially diverse patients in an online format [27,28]. Engaging key stakeholders from the community can serve as a useful tool for answering these questions.

Stakeholder engagement is beneficial for chronic pain patients as it gives them an opportunity to be included in the development of interventions, particularly those aimed to improve their emotional and physical well-being [5,18]. There is limited data available on the stakeholder engagement process for integrative medicine approaches to care, such as mindfulness interventions. Previous reports of stakeholder engagement processes in integrative medicine have focused largely on engaging patients or healthcare providers [29,30]. However, our previous work has identified multi-level implementation determinants relevant to integrative medicine approaches [31] and group medical visits for patients with chronic pain [32]. This protocol is unique in providing multiple methods of engagement and input of community members who represent individual, community, and policy levels of influence in an ongoing trial.

Pragmatic clinical trials both impact and provide valuable benefits to stakeholders at multiple levels of the health system. Patients receive access to evidence-based treatment for cLBP that wouldn't otherwise be available at no cost. Providers can refer their patients to receive services that wouldn't otherwise be available. Healthcare administrators can learn about new treatment options before they become standard of care. Because pragmatic trials are embedded in health systems with multiple stakeholders, it is also crucial to include timely feedback from representatives from those groups, ideally at every phase of the trial.

Principles of community-based participatory research indicate that community input should ideally be included throughout the research lifecycle, from formulating the research question to disseminating findings and acting on next steps [1]. This protocol is limited in that it will not be implemented until the third year of a five-year trial, due to the availability of funding. Ideally, community members would be involved earlier in the research process, helping to design the research questions, study design, selection of measures, and selection of study sites. This work is also limited in that we are only conducting interviews with persons with cLBP who enroll in the OPTIMUM trial and are not engaging with a general

population of patients with cLBP. There is a risk that we will not achieve thematic saturation in either focus groups or individual interviews. To mitigate this risk, we will conduct rapid qualitative analysis to assess themes as they arise and continue with data collection as needed. Further, stakeholder engagement work can be relatively burdensome for participants and researchers. This work can be costly and requires additional paperwork, infrastructure, and time to conduct, potentially slowing the research process. This protocol is also limited in that we are not evaluating the impact of our stakeholder engagement activities outside of qualitative analysis of interviews and cataloging changes made to the implementation of the trial.

A challenge of convening an engaged CAB is recruiting a representative body of stakeholders. Representativeness must be balanced with feasibility, and any group of ten individuals will likely be missing valuable perspectives. However, larger groups meeting regularly may be more difficult to coordinate, and to ensure that all voices are heard at each meeting.

In conclusion, this stakeholder engagement protocol includes multiple data collection methods, such as engaging a community advisory board, conducting focus groups with trial participants, and interviewing multi-level stakeholders to gather input from a diversity of perspectives. We will focus on feedback on both how the trial is being conducted as well as input for future research. Using rapid qualitative analysis methods, including lightning reports and summaries, we will identify and report action items to the multi-site research team in an efficient way. This will allow our team to be responsive to the feedback and suggestions we receive, and to track progress. The intention of using this methodology is to both improve the trial and build trust with stakeholders. Additional thematic content analysis of qualitative data will allow for more detailed analysis and identification of themes across data sources. In this manner, both real time feedback and additional qualitative data will inform how this pragmatic clinical trial is conducted while sowing seeds for future research and implementation collaborations.

## Supporting information

**S1 Appendix. Appendix A:** IRB Approved Protocol Language. The following section contains language approved by the University of Pittsburgh's Institutional Review Board for the Stakeholder Engagement Protocol. **Appendix B:** Interview and Focus Group Guides. The following section contains semi-structured interview and focus group guides for the qualitative portions of the Stakeholder Engagement Protocol.
(DOCX)

## Acknowledgments

The content is solely the responsibility of the authors and does not necessarily represent the official views of the NCCIH or the NIH or its HEAL initiative.

## Author contributions

**Conceptualization:** Isabel J. Roth, Christine R. Lathren, Jessica L. Barnhill, Natalia E. Morone.

**Data curation:** Isabel J. Roth, Elondra D Harr, Ruth D Rodriguez, Jose E Baez.

**Formal analysis:** Isabel J. Roth, Elondra D Harr, Christine R. Lathren, Jessica L. Barnhill, Jose E Baez.

**Funding acquisition:** Isabel J. Roth, Jessica L. Barnhill, Natalia E. Morone.

**Investigation:** Isabel J. Roth, Natalia E. Morone.

**Methodology:** Isabel J. Roth, Christine R. Lathren, Jessica L. Barnhill.

**Project administration:** Elondra D Harr, Ruth D Rodriguez.

**Supervision:** Isabel J. Roth, Natalia E. Morone.

**Validation:** Isabel J. Roth.

**Visualization:** Isabel J. Roth.

**Writing – original draft:** Isabel J. Roth, Elondra D Harr, Christine R. Lathren.

**Writing – review & editing:** Isabel J. Roth, Elondra D Harr, Christine R. Lathren, Jessica L. Barnhill, Ruth D Rodriguez, Jose E Baez, Natalia E. Morone.

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
