## [Decision Letter · Decision Letter 0]

26 Nov 2024

Dear Dr. Roth,

We look forward to receiving your revised manuscript.

Kind regards,

Jan Christopher Cwik, Ph.D.

Academic Editor

PLOS ONE

Journal Requirements:

3. Thank you for stating the following financial disclosure: [This work was supported within the National Institutes of Health (NIH) Pragmatic Trials Collaboratory through the NIH HEAL Initiative under award number UH3AT010621 administered by the National Center for Complementary and Integrative Health (NCCIH). This work also received logistical and technical support from the PRISM Resource Coordinating Center under award number U24AT010961 from the NIH through the NIH HEAL Initiative. The content is solely the responsibility of the authors and does not necessarily represent the official views of the NCCIH or the NIH or its HEAL initiative].

4. Thank you for stating the following in the Acknowledgments Section of your manuscript: [This work was supported within the National Institutes of Health (NIH) Pragmatic Trials Collaboratory through the NIH HEAL Initiative under award number UH3AT010621 administered by the National Center for Complementary and Integrative Health (NCCIH). This work also received logistical and technical support from the PRISM Resource Coordinating Center under award number U24AT010961 from the NIH through the NIH HEAL Initiative. The content is solely the responsibility of the authors and does not necessarily represent the official views of the NCCIH or the NIH or its HEAL initiative.]

Please remove any funding-related text from the manuscript and let us know how you would like to update your Funding Statement. Currently, your Funding Statement reads as follows: [This work was supported within the National Institutes of Health (NIH) Pragmatic Trials Collaboratory through the NIH HEAL Initiative under award number UH3AT010621 administered by the National Center for Complementary and Integrative Health (NCCIH). This work also received logistical and technical support from the PRISM Resource Coordinating Center under award number U24AT010961 from the NIH through the NIH HEAL Initiative. The content is solely the responsibility of the authors and does not necessarily represent the official views of the NCCIH or the NIH or its HEAL initiative].

Reviewers' comments:

Reviewer's Responses to Questions

**Comments to the Author**

1. Does the manuscript provide a valid rationale for the proposed study, with clearly identified and justified research questions?

Reviewer #1: Partly

Reviewer #2: Yes

Reviewer #3: Partly

Reviewer #4: Yes

2. Is the protocol technically sound and planned in a manner that will lead to a meaningful outcome and allow testing the stated hypotheses?

Reviewer #1: Partly

Reviewer #2: Yes

Reviewer #3: Partly

Reviewer #4: Yes

3. Is the methodology feasible and described in sufficient detail to allow the work to be replicable?

Reviewer #1: No

Reviewer #2: Yes

Reviewer #3: Yes

Reviewer #4: Yes

4. Have the authors described where all data underlying the findings will be made available when the study is complete?

Reviewer #1: Yes

Reviewer #2: Yes

Reviewer #3: No

Reviewer #4: Yes

5. Is the manuscript presented in an intelligible fashion and written in standard English?

Reviewer #1: Yes

Reviewer #2: Yes

Reviewer #3: Yes

Reviewer #4: Yes

You may also provide optional suggestions and comments to authors that they might find helpful in planning their study.

Reviewer #1: Background:

Although the authors acknowledge the need for more research and data on the stakeholder engagement process for integrative medicine (p. 13), it is less clear how this protocol will address this knowledge gap. For example, it is not entirely clear if the use of rapid qualitative methods will yield information that can inform how researchers and study teams adapt or change how stakeholders are involved in the design, implementation, or evaluation of pragmatic clinical trials. A more explicit justification about why the authors anticipate that rapid qualitative methods can provide new insight into stakeholder engagement would be beneficial.

Methodology:

Given the parent study is concluding this calendar year, I have some concerns about the lack of details provided by the authors if they do not get adequate representation from patients and stakeholders in the focus groups and interviews they plan to conduct. The emphasis on snowball sampling may be the most appropriate option in this supplement study. Still, there are potential risks that the authors should acknowledge if they do not reach thematic saturation. Specifically, suppose the focus is to enhance the representation of underrepresented patients in the parent study. In that case, it is unclear how the authors will be able to recruit underrepresented patients for the supplement study.

Furthermore, there is a risk of self-selection bias if the authors can only recruit and enroll non-minority patients in their supplement study to understand better the barriers to participating in the parent study. Similarly, on p. 6, the authors state they will invite participants of the parent study to participate in the supplement study focus groups. Without understanding the perspective of eligible participants who did not consent to the parent study, I have reservations about whether the findings of the supplement study will be able to shed light on the factors impeding patients who are less motivated or less likely to engage in trials or research, in general.

Relatedly, the protocol paper lacks sufficient detail to explain how the study team will assess the impact or effectiveness of the stakeholder engagement process. Although qualitative data is necessary to understand the factors inhibiting or enabling patient enrollment, I believe the authors can explain more precisely how they will measure the impact of their community-based participatory research activities.

From a methodological perspective (p. 6), it would be insightful if the authors provided an appendix of the approved interview and focus group guides. How will these guides be modified to accommodate the usual care and intervention groups? Also, on p. 6, what is a policy influencer? Can the authors provide an example of one?

Page 7: What is the justification for including local insurers in the recruitment, retention, integration, and dissemination of MBSR? Please expand.

Also, why will “most interviews” occur virtually? Will the ones conducted in person be audio-recorded, too?

Minor comment: The 3-hour training session with the CAB members sounds interesting. Do the authors anticipate that this type of training would be beneficial to implement more broadly if they anticipate scaling this trial to other sites? Are the authors measuring CAB members’ participation in and satisfaction with this training?

Page 8: The CAB includes various representatives, but I believe an essential group is missing from the protocol: research associates and coordinators of the parent study. Given their central role in the recruitment and retention processes for clinical trials and knowledge of why patients decline to participate, I was hoping the authors could shed light on the decision not to include these roles in their supplement study. Are these types of roles not involved in the parent study?

Finally, how will the authors consolidate the recommendations from the stakeholders and patients to understand what recommendations are most appropriate and actionable? Using a grid that lists each discussion point (p. 10) is a starting point, but how will the CAB members, in conjunction with the study team, identify common recommendations across the three sites that are appropriate (i.e., context-specific) for all study locations? Perhaps the authors may want to consider having participants and study team members rank the recommendations according to the study site so that the recommendations selected for implementation could be specific to each site.

Study status and timeline:

Concerning the current study status and timeline (p. 11), it would be helpful if the authors could expand briefly on the status of their supplement study. Understanding this is a protocol paper; I was still left wanting to know more about the current data collection status of the supplement study. How many focus groups have been completed since the supplement study began in September 2021? How many interviews have been conducted at this point in the supplement study? Has there been any improvement in recruitment for the parent study because of this stakeholder engagement process?

Reviewer #2: O manuscrito inteligível, fornece uma justificativa válida para o estudo proposto, com questões de pesquisa claramente identificadas e justificadas. Metodologia viável. Descreve os métodos em detalhes suficientes

Reviewer #3: Outstanding study and well done. This paper has a potential to be accepted, but some important points have to be clarified or fixed before we can proceed and a positive action can be taken. Given these shortcomings the manuscript requires major revisions. I have some suggestions for each aspect of the manuscript. We here summarize these points:

Abstract:

Please focus the introduction in the abstract on the current study and your aims. The introduction is lengthy, and the last two lines are unnecessary and vague. In the methodology should be summarized, emphasizing the sample of this study and the methods of data collection.

The conclusion provides a clear and structured response to the main question or Purpose of the current research. Please conclude in relation to the overall Purpose or main question of the study and avoid generalizations.

Introduction:

In the introduction section authors should begin with a general context, narrowing to the specific focus of the manuscript. Include answer five main question: why your research is important, what is already known about the topic, the “gap” or what is not yet known about the topic, why it is important to learn the new information that your research adds, and the specific research aim(s) that your manuscript addresses. Your research aim should address the gap you identified. Be sure to add enough background information to enable readers to understand your study. Please clearly demonstrate the research problem of the present study from both a tactical (theoretical necessity) and a content (practical necessity) perspective.

In the introduction, more attention should be given to background information, research problem, research objective(s) and research question, making them tangible and understandable for the reader.

Methodology:

There is very little info on the development of this protocol (details, implementation, fidelity). The explanations are general; please address the details of the design and feasibility of the protocol.

It is recommended to evaluate the psychometric characteristics of the designed protocol in this study based on the principles of developing therapeutic interventions according to the American Psychological Association’s guidelines for assessing treatment protocols in three areas: the process of developing guidelines, Treatment Efficacy, and Clinical Utility.

The first criterion in the process of developing guidelines pertains to the members of the intervention development working group. What areas of expertise did the working group for the designed protocol include?

As the second criterion, the ethical evaluation of the developed program was conducted by whom?

The evaluation of treatment protocols by the American Psychological Association in the section on the therapeutic effectiveness of the intervention includes five criteria. The first criterion pertains to the theoretical foundations of the designed intervention. The second criterion is based on scientific clinical observations and the consensus of experts regarding the intervention topic. The third criterion addresses the effectiveness of the intervention in terms of the desirability of outcomes both exclusively and in comparison, to other related interventions. The fourth criterion in the area of treatment effectiveness relates to the alignment between the treatment and the patient. In the fifth criterion, provide the necessary documentation to specify the outcomes of the intervention and present supporting evidence.

what did the sessions involve, who delivered the sessions, who was present during the session, what format were the sessions delivered in (face-to-face, website, workbooks?), where were the sessions delivered (clinic, home?) A table with the topics of each session is not enough.

Results:

How were the psychometric characteristics of the current protocol examined? How many members were in the focus group, and what specialties did they have? How was the content of the interviews coded? What software was used for coding? Please provide the necessary explanations on these matters.

Qualitative research software should be used for coding interviews

Discussions:

in the discussion should beginning with interpretation of your results and moving to general implications. begins with a restatement of the main findings, which can usually be accomplished with a few carefully-crafted sentences. Next, interpret the meaning or explain the significance of your results, lifting the reader’s gaze from the study’s specific findings to more general applications. Then, compare these study findings with other research. Are these findings in agreement or disagreement with those from other studies? Does this study impart additional nuance to well-accepted theories? Situate your findings within the broader context of scientific literature, then explain the pathways or mechanisms that might give rise to, or explain, the results. Keep in mind that every study has strengths and limitations. Candidly reporting yours helps readers to correctly interpret your research findings. The next element of the discussion is a summary of the potential impacts and applications of the research. Should these results be used to optimally design an intervention? Does the work have implications for clinical protocols or public policy? These considerations will help the reader to further grasp the possible impacts of the presented work. Finally, the discussion should conclude with specific suggestions for future work. Here, you have an opportunity to illuminate specific gaps in the literature that compel further study. Avoid the phrase “future research is necessary” because the recommendation is too general to be helpful to readers. Instead, provide substantive and specific recommendations for future studies.

Conclusions:

Please provide a clear, objective, and tangible response to the overall research question and the feasibility of the current protocol for the target population.

References:

Please use more recent and high-quality references.

The text of the article needs to be simplified and made more fluid for better communication with the reader. Therefore, it is suggested that the authors simplify, trim, and summarize some of the long and unnecessary sentences to keep the reader’s mind aligned with the flow of the study’s progression.

Reviewer #4: This is an excellent piece of writing title “A Protocol for Using Rapid Qualitative Techniques to Incorporate Multi-Level Stakeholder Feedback in a Pragmatic Clinical Trial of Mindfulness for Chronic Low Back Pain”. However, there are few things which need clarification of the methods to diagnose low back pain. What tools would be used to rule out the diagnosis? Whether it would be a categorical or dimensional classification?

As the intervention would be given as part of tele health intervention, authors need to give details that how the retention would be managed. Further, authors need to explain the critical incidents during the intervention phase, how you would address the issues, and any escalation plan in hand to address the distress of patients experienced during the intervention.

Authors have mentioned that Individuals such as “clinic staff (front desk/registration, nurses, nurse practitioners, medical assistants, clinic managers, medical directors), healthcare system professionals (medical staff directors, providers, nursing staff directors, clinic and health system administrators), and payors (insurance representatives) will be providing their expert opinions in one-on-one interviews with trained study personnel”. Further they will undergo verbal informed consent, any reason for not taking written informed consent, give some reasons for this.

**Do you want your identity to be public for this peer review?** For information about this choice, including consent withdrawal, please see our Privacy Policy

Reviewer #1: No

Reviewer #2: **Yes: ** Ana Carina Henriques Teodósio Moisão

Reviewer #3: No

Reviewer #4: **Yes: ** Salman Shahzad

---

## [Author Response · Author response to Decision Letter 1]

24 Jan 2025

Reviewer #1: Background:

Although the authors acknowledge the need for more research and data on the stakeholder engagement process for integrative medicine (p. 13), it is less clear how this protocol will address this knowledge gap. For example, it is not entirely clear if the use of rapid qualitative methods will yield information that can inform how researchers and study teams adapt or change how stakeholders are involved in the design, implementation, or evaluation of pragmatic clinical trials. A more explicit justification about why the authors anticipate that rapid qualitative methods can provide new insight into stakeholder engagement would be beneficial.

Thank you for this comment. We have added additional references about rapid qualitative methods being useful for incorporating stakeholder input, as well as explaining that rapid qualitative methods allow us to incorporate feedback in real time (lines 84-85).

Methodology:

Given the parent study is concluding this calendar year, I have some concerns about the lack of details provided by the authors if they do not get adequate representation from patients and stakeholders in the focus groups and interviews they plan to conduct. The emphasis on snowball sampling may be the most appropriate option in this supplement study. Still, there are potential risks that the authors should acknowledge if they do not reach thematic saturation. Specifically, suppose the focus is to enhance the representation of underrepresented patients in the parent study. In that case, it is unclear how the authors will be able to recruit underrepresented patients for the supplement study.

We have added additional detail on the limitations of our sampling strategies and potential for not reaching thematic saturation in the discussion section (lines 257-271). Unfortunately, due to the timing of the review process, we are not able to change the methodology at this time.

The focus is to enhance representation of patient voices into the parent trial, therefore, the representation of patients will be inherently limited by the recruited patients through the parent trial.

Furthermore, there is a risk of self-selection bias if the authors can only recruit and enroll non-minority patients in their supplement study to understand better the barriers to participating in the parent study. Similarly, on p. 6, the authors state they will invite participants of the parent study to participate in the supplement study focus groups. Without understanding the perspective of eligible participants who did not consent to the parent study, I have reservations about whether the findings of the supplement study will be able to shed light on the factors impeding patients who are less motivated or less likely to engage in trials or research, in general.

Our IRB requires us to delete the information of people who decline participation, it is therefore considered unethical to re-contact these participants after they have declined participation. We have address this as a limitation in the discussion section (lines 263-264).

Relatedly, the protocol paper lacks sufficient detail to explain how the study team will assess the impact or effectiveness of the stakeholder engagement process. Although qualitative data is necessary to understand the factors inhibiting or enabling patient enrollment, I believe the authors can explain more precisely how they will measure the impact of their community-based participatory research activities.

We have acknowledged the limitations of our our qualitative analysis and cataloging of stakeholder-recommended changes and actions taken in the discussion section (lines 313-315).

From a methodological perspective (p. 6), it would be insightful if the authors provided an appendix of the approved interview and focus group guides.

We have included these guides as Appendix B.

How will these guides be modified to accommodate the usual care and intervention groups?

There are separate interview guides for the usual care and intervention focus groups. This has been clarified on lines 129-131.

Also, on p. 6, what is a policy influencer? Can the authors provide an example of one?

Clarify advocacy groups

Page 7: What is the justification for including local insurers in the recruitment, retention, integration, and dissemination of MBSR? Please expand.

Understand how local insurers make decisions about coverage

Also, why will “most interviews” occur virtually? Will the ones conducted in person be audio-recorded, too?

We have deleted the word most to reflect that all interviews are conducted virtually (line 150).

Minor comment: The 3-hour training session with the CAB members sounds interesting. Do the authors anticipate that this type of training would be beneficial to implement more broadly if they anticipate scaling this trial to other sites? Are the authors measuring CAB members’ participation in and satisfaction with this training?

Gave verbal feedback

Page 8: The CAB includes various representatives, but I believe an essential group is missing from the protocol: research associates and coordinators of the parent study. Given their central role in the recruitment and retention processes for clinical trials and knowledge of why patients decline to participate, I was hoping the authors could shed light on the decision not to include these roles in their supplement study. Are these types of roles not involved in the parent study?

Included

Finally, how will the authors consolidate the recommendations from the stakeholders and patients to understand what recommendations are most appropriate and actionable? Using a grid that lists each discussion point (p. 10) is a starting point, but how will the CAB members, in conjunction with the study team, identify common recommendations across the three sites that are appropriate (i.e., context-specific) for all study locations? Perhaps the authors may want to consider having participants and study team members rank the recommendations according to the study site so that the recommendations selected for implementation could be specific to each site.

The idea of including ranking is a good one, however due to the timing of the review process, we are unable to change the protocol itself at this time.

Study status and timeline:

Concerning the current study status and timeline (p. 11), it would be helpful if the authors could expand briefly on the status of their supplement study. Understanding this is a protocol paper; I was still left wanting to know more about the current data collection status of the supplement study. How many focus groups have been completed since the supplement study began in September 2021? How many interviews have been conducted at this point in the supplement study? Has there been any improvement in recruitment for the parent study because of this stakeholder engagement process?

Update on recruitment

These details will be reported in the outcomes manuscript. Following convention, this manuscript only pertains to the protocol.

Reviewer #2: O manuscrito inteligível, fornece uma justificativa válida para o estudo proposto, com questões de pesquisa claramente identificadas e justificadas. Metodologia viável. Descreve os métodos em detalhes suficientes

Thank you

Outstanding study and well done. This paper has a potential to be accepted, but some important points have to be clarified or fixed before we can proceed and a positive action can be taken. Given these shortcomings the manuscript requires major revisions. I have some suggestions for each aspect of the manuscript. We here summarize these points:

Abstract:

Please focus the introduction in the abstract on the current study and your aims. The introduction is lengthy, and the last two lines are unnecessary and vague. 

Clarified aims within introduction, including clarifying last two lines.

In the methodology should be summarized, emphasizing the sample of this study and the methods of data collection.

Clarified language to increase readability.

The conclusion provides a clear and structured response to the main question or Purpose of the current research. Please conclude in relation to the overall Purpose or main question of the study and avoid generalizations.

Introduction:

In the introduction section authors should begin with a general context, narrowing to the specific focus of the manuscript.

Include answer five main question: why your research is important, what is already known about the topic, the “gap” or what is not yet known about the topic, why it is important to learn the new information that your research adds, and the specific research aim(s) that your manuscript addresses. Your research aim should address the gap you identified. Be sure to add enough background information to enable readers to understand your study. Please clearly demonstrate the research problem of the present study from both a tactical (theoretical necessity) and a content (practical necessity) perspective.

In the introduction, more attention should be given to background information, research problem, research objective(s) and research question, making them tangible and understandable for the reader.

Methodology:

There is very little info on the development of this protocol (details, implementation, fidelity). The explanations are general; please address the details of the design and feasibility of the protocol.

It is recommended to evaluate the psychometric characteristics of the designed protocol in this study based on the principles of developing therapeutic interventions according to the American Psychological Association’s guidelines for assessing treatment protocols in three areas:

the process of developing guidelines, Treatment Efficacy,

and Clinical Utility.

The first criterion in the process of developing guidelines pertains to the members of the intervention development working group. What areas of expertise did the working group for the designed protocol include?

As the second criterion, the ethical evaluation of the developed program was conducted by whom?

The evaluation of treatment protocols by the American Psychological Association in the section on the therapeutic effectiveness of the intervention includes five criteria. The first criterion pertains to the theoretical foundations of the designed intervention. The second criterion is based on scientific clinical observations and the consensus of experts regarding the intervention topic. The third criterion addresses the effectiveness of the intervention in terms of the desirability of outcomes both exclusively and in comparison, to other related interventions. The fourth criterion in the area of treatment effectiveness relates to the alignment between the treatment and the patient. In the fifth criterion, provide the necessary documentation to specify the outcomes of the intervention and present supporting evidence.

Thank you for this information. The protocol presented here is specific to the inclusion of stakeholder input in a clinical trial, but this protocol is not about the treatment protocol itself. For more details on the treatment protocol, please see: Greco CM, Gaylord SA, Faurot K, et al. The design and methods of the OPTIMUM study: A multisite pragmatic randomized clinical trial of a telehealth group mindfulness program for persons with chronic low back pain. Contemp Clin Trials. 2021;109:106545. doi:10.1016/j.cct.2021.106545

what did the sessions involve, who delivered the sessions, who was present during the session, what format were the sessions delivered in (face-to-face, website, workbooks?), where were the sessions delivered (clinic, home?) A table with the topics of each session is not enough.

Community Advisory Board (CAB) meetings include all members of the CAB (detailed on p. #) as well as relevant study team members…

Results:

How were the psychometric characteristics of the current protocol examined? How many members were in the focus group, and what specialties did they have? How was the content of the interviews coded? What software was used for coding? Please provide the necessary explanations on these matters.

Qualitative research software should be used for coding interviews

The intent of this protocol was not to measure psychometric characteristics. Psychometrics are therefore not discussed in this manuscript.

Focus groups will contain about 8 participants (line 133), and will be constituted of members of the control and intervention groups from the parent trial. Analytic approaches (including coding and data management) are detailed on lines 166-178. We have clarified that ATLAS.ti is the qualitative data management software package we are using (line 175)

Discussions:

in the discussion should beginning with interpretation of your results and moving to general implications. begins with a restatement of the main findings, which can usually be accomplished with a few carefully-crafted sentences. Next, interpret the meaning or explain the significance of your results, lifting the reader’s gaze from the study’s specific findings to more general applications. Then, compare these study findings with other research. Are these findings in agreement or disagreement with those from other studies? Does this study impart additional nuance to well-accepted theories? Situate your findings within the broader context of scientific literature, then explain the pathways or mechanisms that might give rise to, or explain, the results. Keep in mind that every study has strengths and limitations. Candidly reporting yours helps readers to correctly interpret your research findings. The next element of the discussion is a summary of the potential impacts and applications of the research. Should these results be used to optimally design an intervention? Does the work have implications for clinical protocols or public policy? These considerations will help the reader to further grasp the possible impacts of the presented work. Finally, the discussion should conclude with specific suggestions for future work. Here, you have an opportunity to illuminate specific gaps in the literature that compel further study. Avoid the phrase “future research is necessary” because the recommendation is too general to be helpful to readers. Instead, provide substantive and specific recommendations for future studies.

Thank you for this information. Because this manuscript details a protocol, we did not include results or interpretation of findings in the discussion section.

Conclusions:

Please provide a clear, objective, and tangible response to the overall research question and the feasibility of the current protocol for the target population.

References:

Please use more recent and high-quality references.

We have used methodological references that we felt were appropriate for describing our methodolofy.

The text of the article needs to be simplified and made more fluid for better communication with the reader. Therefore, it is suggested that the authors simplify, trim, and summarize some of the long and unnecessary sentences to keep the reader’s mind aligned with the flow of the study’s progression.

Thank you for these suggestions. We have revised to improve readability.

Reviewer #4: This is an excellent piece of writing title “A Protocol for Using Rapid Qualitative Techniques to Incorporate Multi-Level Stakeholder Feedback in a Pragmatic Clinical Trial of Mindfulness for Chronic Low Back Pain”. However, there are few things which need clarification of the methods to diagnose low back pain. What tools would be used to rule out the diagnosis? Whether it would be a categorical or dimensional classification?

As the intervention would be given as part of tele health intervention, authors need to give details that how the retention would be managed. Further, authors need to explain the critical incidents during the intervention phase, how you would address the issues, and any escalation plan in hand to address the distress of patients

---

## [Decision Letter · Decision Letter 1]

17 Apr 2025

PLOS ONE

Dear Dr. Roth,

Thank you for submitting your manuscript to PLOS ONE. After careful consideration, we feel that it has merit but does not fully meet PLOS ONE’s publication criteria as it currently stands. Therefore, we invite you to submit a revised version of the manuscript that addresses the points raised during the review process.

We look forward to receiving your revised manuscript.

Kind regards,

Jan Christopher Cwik, Ph.D.

Academic Editor

PLOS ONE

Journal Requirements:

Additional Editor Comments:

Thank you for submitting your revised manuscript. I have now received feedback from the reviewers on your revisions. Two of the reviewers have noted some minor aspects, which I would like you to consider adjusting in a further revision.

Reviewers' comments:

Reviewer's Responses to Questions

**Comments to the Author**

1. Does the manuscript provide a valid rationale for the proposed study, with clearly identified and justified research questions?

Reviewer #1: Yes

Reviewer #2: Yes

Reviewer #3: Yes

2. Is the protocol technically sound and planned in a manner that will lead to a meaningful outcome and allow testing the stated hypotheses?

Reviewer #1: Yes

Reviewer #2: Partly

Reviewer #3: Yes

3. Is the methodology feasible and described in sufficient detail to allow the work to be replicable?

Reviewer #1: No

Reviewer #2: Yes

Reviewer #3: Yes

4. Have the authors described where all data underlying the findings will be made available when the study is complete?

Reviewer #1: Yes

Reviewer #2: Yes

Reviewer #3: Yes

5. Is the manuscript presented in an intelligible fashion and written in standard English?

Reviewer #1: Yes

Reviewer #2: Yes

Reviewer #3: Yes

You may also provide optional suggestions and comments to authors that they might find helpful in planning their study.

Reviewer #1: Thank you for responding to the reviewers' comments. However, some unresolved issues remain. I cannot locate Appendix B in the revision. What new references about rapid qualitative methods were added to the revised manuscript? If policy influencer and advocacy groups are used synonymously throughout the paper, it would be beneficial to use only one of those terms.

Reviewer #2: I agree with the reviewer #1 :Although the authors acknowledge the need for more research and data on the stakeholder engagement process for integrative medicine, it is less clear how this protocol will address this knowledge gap.

Given the parent study is concluding this calendar year, I have some concerns about the lack of details provided by the authors if they do not get adequate representation from patients and stakeholders in the focus groups and interviews they plan to conduct. The emphasis on snowball sampling may be the most appropriate option in this supplement study. Still, there are potential risks that the authors should acknowledge if they do not reach thematic saturation. Specifically, suppose the focus is to enhance the representation of underrepresented patients in the parent study. In that case, it is unclear how the authors will be able to recruit underrepresented patients for the supplement study.

Reviewer #3: Given that the authors have thoroughly addressed all requested comments and effectively resolved the raised concerns, and that the manuscript now meets the requisite criteria for publication, I find no specific issues to highlight. Therefore, I conclude that the paper aligns well with the journal’s established standards for quality and rigor.

**Do you want your identity to be public for this peer review?** For information about this choice, including consent withdrawal, please see our Privacy Policy

Reviewer #1: No

Reviewer #2: No

Reviewer #3: No

---

## [Author Response · Author response to Decision Letter 2]

24 Jun 2025

Reviewer #1:  Thank you for responding to the reviewers' comments. However, some unresolved issues remain.

I cannot locate Appendix B in the revision.

What new references about rapid qualitative methods were added to the revised manuscript?

If policy influencer and advocacy groups are used synonymously throughout the paper, it would be beneficial to use only one of those terms.

Thank you for this comment.

Appendix B was previously attached as a separate document, we have also added Appendix B to the end of the manuscript file for ease of review for this submission.

New references about rapid qualitative methods include:

[8] R. E. Keith, J. C. Crosson, A. S. O’Malley, D. Cromp, and E. F. Taylor, “Using the Consolidated Framework for Implementation Research (CFIR) to produce actionable findings: a rapid-cycle evaluation approach to improving implementation.,” Implement. Sci., vol. 12, no. 1, p. 15, Feb. 2017, doi: 10.1186/s13012-017-0550-7.

[9] C. Vindrola-Padros and G. A. Johnson, “Rapid techniques in qualitative research: A critical review of the literature.,” Qual. Health Res., vol. 30, no. 10, pp. 1596–1604, Aug. 2020, doi: 10.1177/1049732320921835.

[10] B. Taylor, C. Henshall, S. Kenyon, I. Litchfield, and S. Greenfield, “Can rapid approaches to qualitative analysis deliver timely, valid findings to clinical leaders? A mixed methods study comparing rapid and thematic analysis.,” BMJ Open, vol. 8, no. 10, p. e019993, Oct. 2018, doi: 10.1136/bmjopen-2017-019993.

[11] A. L. Nevedal et al., “Rapid versus traditional qualitative analysis using the Consolidated Framework for Implementation Research (CFIR).,” Implement. Sci., vol. 16, no. 1, p. 67, Jul. 2021, doi: 10.1186/s13012-021-01111-5.

We have changed the terminology throughout the document to refer to ‘policy advocates’

Reviewer #2: I agree with the reviewer #1 :”Although the authors acknowledge the need for more research and data on the stakeholder engagement process for integrative medicine, it is less clear how this protocol will address this knowledge gap.

Given the parent study is concluding this calendar year, I have some concerns about the lack of details provided by the authors if they do not get adequate representation from patients and stakeholders in the focus groups and interviews they plan to conduct. The emphasis on snowball sampling may be the most appropriate option in this supplement study. Still, there are potential risks that the authors should acknowledge if they do not reach thematic saturation. Specifically, suppose the focus is to enhance the representation of underrepresented patients in the parent study. In that case, it is unclear how the authors will be able to recruit underrepresented patients for the supplement study.”

Thank you for this comment. We have added additional references about rapid qualitative methods being useful for incorporating stakeholder input, as well as explaining that rapid qualitative methods allow us to incorporate feedback in real time (lines 59-60).

We added additional detail on the limitations of our sampling strategies and potential for not reaching thematic saturation in the discussion section in our previous revision, based on reviewer 1’s feedback (lines 257-271). Unfortunately, due to the timing of the review process, we are not able to change the methodology at this time.

The focus is to enhance representation of patient voices into the parent trial, therefore, the representation of patients will be inherently limited by the recruited patients through the parent trial.

---

## [Decision Letter · Decision Letter 2]

5 Aug 2025

Dear Dr. Roth,

Thank you for submitting your manuscript to PLOS ONE. After careful consideration, we feel that it has merit but does not fully meet PLOS ONE’s publication criteria as it currently stands. Therefore, we invite you to submit a revised version of the manuscript that addresses the points raised during the review process.

We look forward to receiving your revised manuscript.

Kind regards,

Jan Christopher Cwik, Prof. Dr. Dr.

Academic Editor

PLOS ONE

Journal Requirements:

Additional Editor Comments:

Dear authors,

As you can see from the attached feedback, one reviewer has requested some revisions. I would therefore like to ask you to adapt these in a further revision process of your manuscript.

Reviewers' comments:

Reviewer's Responses to Questions

**Comments to the Author**

1. Does the manuscript provide a valid rationale for the proposed study, with clearly identified and justified research questions?

Reviewer #1: Yes

Reviewer #2: Yes

2. Is the protocol technically sound and planned in a manner that will lead to a meaningful outcome and allow testing the stated hypotheses?

Reviewer #1: Yes

Reviewer #2: Yes

3. Is the methodology feasible and described in sufficient detail to allow the work to be replicable?

Reviewer #1: Yes

Reviewer #2: Yes

4. Have the authors described where all data underlying the findings will be made available when the study is complete?

Reviewer #1: Yes

Reviewer #2: Yes

5. Is the manuscript presented in an intelligible fashion and written in standard English?

Reviewer #1: Yes

Reviewer #2: Yes

You may also provide optional suggestions and comments to authors that they might find helpful in planning their study.

Reviewer #1: I do not have any additional comments to provide to the authors. Thank you for providing the interview guides.

Reviewer #2: It is unclear how the authors will be able to recruit underrepresented patients for

the supplement study.

Here are potential risks that the authors should acknowledge if they do not reach thematic saturation

**Do you want your identity to be public for this peer review?** For information about this choice, including consent withdrawal, please see our Privacy Policy

Reviewer #1: No

Reviewer #2: **Yes: ** Ana Carina Henriques Teodósio Moisão

---

## [Author Response · Author response to Decision Letter 3]

15 Sep 2025

REVIEWER: It is unclear how the authors will be able to recruit underrepresented patients for

the supplement study.

RESPONSE: We have included a detailed explanation of our sampling strategy to ensure a diverse sample of underrepresented demographic groups on page 7, lines 142-49:

“To ensure that a diverse sample of underrepresented patients are represented in the focus groups in the supplement study, we will first evaluate the reported demographic factors of all potential participants. A random number generator will be used to select 10 individuals, followed by an assessment to ensure that the group reflects the diversity of the parent study population. Participants are then assigned a number attached to their participant ID and the generator will randomly select a number. The research team will then contact the participant to invite them to participate and if a selected participant is unavailable, another demographically similar participant was contacted to maintain group diversity.“

REVIEWER: Here are potential risks that the authors should acknowledge if they do not reach thematic saturation

RESPONSE: To address risks of not reaching thematic saturation, we have modified the methods on page 7, lines 138-140 and page 8, lines 163-164 to specify that data collection will continue until thematic saturation is reached. We have also acknowledged risks of not reaching thematic saturation in the discussion section on page 15, lines 320-322 and explained how we are mitigating this risk.

---

## [Decision Letter · Decision Letter 3]

21 Nov 2025

A Protocol for Using Rapid Qualitative Techniques to Incorporate Multi-Level Stakeholder Feedback in a Pragmatic Clinical Trial of Mindfulness for Chronic Low Back Pain

PONE-D-23-30839R3

Dear Dr. Roth,

We’re pleased to inform you that your manuscript has been judged scientifically suitable for publication and will be formally accepted for publication once it meets all outstanding technical requirements.

Kind regards,

Jan Christopher Cwik, Prof. Dr. Dr.

Academic Editor

PLOS ONE

Reviewers' comments:

Reviewer's Responses to Questions

**Comments to the Author**

1. Does the manuscript provide a valid rationale for the proposed study, with clearly identified and justified research questions?

Reviewer #2: Yes

2. Is the protocol technically sound and planned in a manner that will lead to a meaningful outcome and allow testing the stated hypotheses?

Reviewer #2: Yes

3. Is the methodology feasible and described in sufficient detail to allow the work to be replicable?

Reviewer #2: Yes

4. Have the authors described where all data underlying the findings will be made available when the study is complete?

Reviewer #2: Yes

5. Is the manuscript presented in an intelligible fashion and written in standard English?

Reviewer #2: Yes

You may also provide optional suggestions and comments to authors that they might find helpful in planning their study.

Reviewer #2: A questão de pesquisa delineada aborda um problema ou tópico académico válido e contribuiu para a base de conhecimento na área.

Descreve os métodos com detalhes suficientes para evitar flexibilidade não divulgada no procedimento experimental ou na linha de análise, incluindo condições neutras em relação ao resultado suficientes para testar as hipóteses propostas e uma análise de poder estatístico.

Descrição de métodos e materiais suficientes

**Do you want your identity to be public for this peer review?** For information about this choice, including consent withdrawal, please see our Privacy Policy

Reviewer #2: **Yes: ** Ana Carina Henriques Teodósio Moisão

---

## [Editor Report · Acceptance letter]

PONE-D-23-30839R3

PLOS One

Dear Dr. Roth,

I'm pleased to inform you that your manuscript has been deemed suitable for publication in PLOS One. Congratulations! Your manuscript is now being handed over to our production team.

Kind regards,

on behalf of

Prof. Dr. Dr. Jan Christopher Cwik

Academic Editor

PLOS One